

# Microglial depletion rescues spatial memory impairment caused by LPS administration in adult mice

Tao Zong[1,2,*], Na Li[2,3,*], Fubing Han[2,4], Junru Liu[5], Mingru Deng[2,5], Vincent Li[6], Meng Zhang[2], Yu Zhou[1,2,5,7] and Ming Yu[7]

[1] Affiliated Qingdao Third People's Hospital, Department of Otorhinolaryngology Head and Neck, Qingdao University, Qingdao, China
[2] Institute of Brain Sciences and Related Disorders, Qingdao University, Qingdao, China, China
[3] Qingdao Binhai University, Qingdao, Shandong, China
[4] Department of Neurosurgery, Affiliated Hospital of Qingdao University, Qingdao, China, China
[5] Department of Neurology, Affiliated Qingdao Central Hospital, University of Health and Rehabilitation Sciences (Qingdao Central Medical Group), Qingdao, China, China
[6] Beverly Hills High School, Unaffiliated, Beverly Hills, California, United States
[7] Department of Health and Life Sciences, University of Health and Rehabilitation Sciences, Qingdao, China
* These authors contributed equally to this work.

Corresponding authors
Yu Zhou, yuzhou@qdu.edu.cn
Ming Yu, ym218@163.com

## ABSTRACT

Recent studies have highlighted the importance of microglia, the resident macrophages in the brain, in regulating cognitive functions such as learning and memory in both healthy and diseased states. However, there are conflicting results and the underlying mechanisms are not fully understood. In this study, we examined the effect of depleting adult microglia on spatial learning and memory under both physiological conditions and lipopolysaccharide (LPS)-induced neuroinflammation. Our results revealed that microglial depletion by PLX5622 caused mild spatial memory impairment in mice under physiological conditions; however, it prevented memory deficits induced by systemic LPS insult. Inactivating microglia through minocycline administration replicated the protective effect of microglial depletion on LPS-induced memory impairment. Furthermore, our study showed that PLX5622 treatment suppressed LPS-induced neuroinflammation, microglial activation, and synaptic dysfunction. These results strengthen the evidence for the involvement of microglial immunoactivation in LPS-induced synaptic and cognitive malfunctions. They also suggest that targeting microglia may be a potential approach to treating neuroinflammation-associated cognitive dysfunction seen in neurodegenerative diseases.

# INTRODUCTION

Microglia are self-renewing, resident immune cells in the central nervous system (CNS) that play essential roles in maintaining brain homeostasis from the embryonic brain rudiment to the aged brain (*Lukens & Eyo, 2022*). Under physiological conditions,

microglia constantly sample the CNS environment, quickly respond to changes, and maintain homeostasis. They are the most dynamic cells in the adult brain that actively regulate spine pruning, neuronal excitability, and blood-brain barrier (BBB) (*Paolicelli et al., 2022*). However, under pathological conditions such as neurodegenerative diseases, microglia become dysfunctional, transitioning from an activated state to a neurotoxic state and contributing to disease progression and cognitive deficit (*Colonna & Butovsky, 2017*).

The functional states of microglia are plastic and strongly depend on context, such as the type and extent of CNS insult or damage (*Hanisch & Kettenmann, 2007*). Traditionally, microglial activation is classified into M1 and M2 states. M1 activation mainly release proinflammatory cytokines such as interleukin-1β (IL-1β), IL-6, and tumor necrosis factor alpha (TNF-α), leading to synaptic dysfunction and eventual neuronal death. In contrast, M2 activation mainly produce anti-inflammatory cytokines such as IL-10 and transforming growth factor beta (TGF-β), playing anti-inflammatory and neuroprotective roles. In the context of chronic neuroinflammation, the balance between pro-inflammatory and anti-inflammatory microglia is disrupted, especially in the hippocampus, resulting in memory impairment (*Cornell et al., 2022*; *Li et al., 2022a*). Recent transcriptome studies have revealed the coexistence of many different activation states of microglia besides the traditional M1/M2 paradigm, which may closely associate with their diverse functions *in vivo* (*Li et al., 2023a*).

Neuroinflammation can be initiated by various stimuli and is a complex combination of acute and chronic inflammatory responses in the CNS (*Sochocka, Diniz & Leszek, 2017*). Chronic neuroinflammation has been considered a common detrimental factor contributing to the pathogenesis of neurodegenerative diseases (*e.g.*, Alzheimer's disease, AD) and neuropsychiatric disorders (*e.g.*, Autism spectrum disorder, ASD) characterized by cognitive disability (*Matta, Hill-Yardin & Crack, 2019*; *Leng & Edison, 2021*). Microglia play a critical role in the onset and progression of neuroinflammation (*Aguzzi, Barres & Bennett, 2013*). Lipopolysaccharides (LPS), found in the outer membrane of gram-negative bacteria, is commonly used as proinflammatory stimuli to induce neuroinflammation both *in vivo* and *in vitro* (*Skrzypczak-Wiercioch & Sałat, 2022*). Studies have shown that both systemic administration (*e.g.*, intraperitoneal, i.p.) and local microinjection (*e.g.*, intracerebroventricular, i.c.v.) of LPS can strongly activate microglia in the brain and lead to cognitive impairments (*Zhao et al., 2019*; *Jung et al., 2023*).

While studies have emphasized the importance of microglia in modulating cognitive functions in both healthy and diseased brains, the results are controversial and the underlying mechanisms are not fully understood. For example, recent studies revealed that microglia support recognition memory (*Basilico et al., 2022b*; *Yegla et al., 2021*), mediate fear memory forgetting (*Wang et al., 2020*), facilitate fear memory extinction (*Yegla et al., 2021*; *Allen et al., 2020*), and regulate spatial memory in healthy rodents (*Elmore et al., 2014*; *Rice et al., 2015*). However, other studies with microglia depletion have suggested that microglia might not play a key role in shaping these cognition-related behaviors (*Allen et al., 2020*; *Elmore et al., 2014*; *De Luca et al., 2020*; *Feng et al., 2017*; *Willis et al., 2020*). Similarly, microglial depletion in rodent models of brain disorders also generates heterogeneous effect on disease-associated cognitive dysfunction. For instance, studies

have reported memory improvement by depletion of microglia in traumatic brain injury mice (*Willis et al., 2020*; *Henry et al., 2020*), rotenone-induced Parkinson's disease (PD) model mice (*Zhang et al., 2021*), and 3xTg-AD model mice (*Dagher et al., 2015*). However, others have found that microglia depletion fails to rescue memory deficit in epilepsy rats (*Wyatt-Johnson et al., 2021*) and APP/PS1-AD model mice (*Unger et al., 2018*), on the contrary, it worsens motor function in MPTP-induced PD model mice (*Yang et al., 2018*). All these contentious findings demonstrate the gap of our knowledge and the need of more studies on microglial function in cognition. To this end, we aim in this study to explore the role of microglia in shaping spatial learning and memory.

The colony-stimulating factor 1 receptor (CSF1R) is crucial for microglial proliferation and survival. The administration of PLX5622, a small molecule inhibitor of CSF1R, was reported to induce a rapid depletion of microglial (*Elmore et al., 2014*; *Erblich et al., 2011*). Minocycline, a tetracycline antibiotic with anti-inflammatory properties, was confirmed to inhibit microglial activation and the release of pro-inflammatory cytokines (*Tikka et al., 2001*; *Šimončičová et al., 2022*). Therefore, we investigated the effect of PLX5622 and minocycline on spatial learning and memory in mice under both physiological conditions and LPS-induced neuroinflammation. Our findings support the dual role of microglia in regulating spatial learning and memory in the adult brain.

## MATERIALS AND METHODS

### Animal care and housing

C57BL/6J mice were chosen for this study due to their well-characterized genetic background, consistent immune response, and susceptibility to LPS-induced neuroinflammation (*da Silva et al., 2024*; *Piirsalu et al., 2022*). We obtained C57BL/6J mice at 8 weeks of age from Vital River Laboratory Animal Technology Co. (Beijing, China). Upon arrival, the animals were acclimatized for 2 weeks before the start of the experiments. Mice were housed in groups of 4 per cage under constant conditions of temperature (21 ± 2 °C) and humidity (50 ± 10%), with a 12-h light/dark cycle, and *ad libitum* access to food and water. All behavioral experiments were conducted using male adult mice (3–4 months old, weighing 25–30 g), during the light cycle from 9:00 am to 6:00 pm. The Chancellor's Animal Research Committee at Qingdao University approved all animal protocols (#3207090367) used in this study, following the National Institutes of Health guidelines.

### LPS administration

Intraperitoneal injections of 0.5 mg/kg LPS (Sigma, Tokyo, Japan) were given daily for 7 days, with the dosage adjusted based on previous studies (*Zhao et al., 2019*). The final LPS injection was done 24 h before the start of behavioral experiments. Injections were performed with a 27-gauge needle in the lower right quadrant of the abdomen to avoid vital organs, using a volume not exceeding 10 mL/kg. Mice were gently restrained during injections to minimize stress, and their condition was closely monitored afterward for any signs of distress or adverse reactions. If any animal showed discomfort, appropriate care was provided, including human euthanasia as per guidelines of the Chancellor's Animal Research Committee at Qingdao University. No anesthesia was given for i.p. injections.

## PLX5622 and minocycline treatment

PLX5622 is a highly selective CSF1R inhibitor known for its effectiveness in depleting microglia (*Spangenberg et al., 2019*). Mice were fed with PLX5622 (1,200 mg/kg) or control chow (Dyets Inc., Bethlehem, PA, USA) for 3 weeks. The 3-week administration period is chosen based on both previous studies (*Badimon et al., 2020*) and our preliminary results, which demonstrate that this duration is minimal to achieve significant microglial depletion without causing adverse effects on overall health. The intraperitoneal injections of 100 mg/kg minocycline (Sigma, Tokyo, Japan) were administered daily for 7 days (*Wang et al., 2020*; *Dagher et al., 2015*). The injection protocol and post-injection monitoring for minocycline were same as for LPS injection.

Different batches of mice were used in the following *in vivo* and *ex vivo* experiments. The hippocampal tissues used in different studies were obtained from separated batches of mice brains. Double-blind experiments were conducted by separate groups of researchers to minimize bias and ensure objective results. Specifically, JL and MD administered the drugs. NL conducted the behavioral and electrophysiological assessments. FH carried out the immunostaining, quantitative reverse transcription PCR, and enzyme-linked immunosorbent assays. Data analysis was handled by TZ, VL and MZ: TZ and VL analyzed the behavioral results, VL analyzed the electrophysiological results, and MZ analyzed the remaining results. NL, FH, TZ, VL and MZ were blinded to the treatment conditions during all assessments and data analyses.

## Behavioral assessments

All behavioral assessments were done according to our previous studies (*Lu et al., 2019*; *Liu et al., 2024*). To evaluate the overall behavioral state of the mice and ensure that differences in learning and memory performance are not attributed to changes in motor function or anxiety levels, we first conducted the Elevated Plus Maze (EPM) and Open Field (OF) tests before cognitive assessment. The EPM consists of a central area, two closed arms with walls (16.5 cm height), and two open arms without walls. Each arm measures 30 cm in length and 6 cm in width, while the main frame stands at a height of 50 cm from the ground. Mice were released from the center and allowed to freely explore the maze for 10 min. The time spent in each arm and total travel distance were analyzed.

The OF test was conducted in a $28 \times 28 \times 35$ cm square arena, divided into a $12 \times 12$ cm center zone and the left periphery zone. Mice were released from the center and allowed to freely explore for 10 min. The total distance traveled and time spent in both zones were analyzed.

The Morris water maze (MWM) circular water pool is 120 cm in diameter and 30 cm deep, divided into four quadrants. Mice undergo training with an invisible platform placed just below the water's surface, aiming to climb onto it to escape the water. Training consists of 4 trials/2 blocks per day for 7 days, with an inter-block interval of 1 h and an inter-trial interval of 30 s. A trial ends when mice reach the platform or after a cut-off time of 60 s. To assess spatial memory, a probe test is conducted 1 h after training on the 3rd, 5th, and 7th days respectively. In these tests, the hidden platform is removed and mice are allowed to navigate the pool for 60 s.

Mice were acclimated to the experimental environment for at least 1 h before behavioral tests. Their behaviors were recorded and analyzed using Noldus EthoVision XT software by two independent researchers unaware of the treatment details.

## Electrophysiological recordings *ex vivo*

Mice were deeply anesthetized with 3% isoflurane and intracardially perfused with saline. Coronal hippocampal slices (300 μm thick) were freshly prepared using a Leica VT-1000 vibratome as described previously (*Cui et al., 2016*). Slices were perfused with 32 °C artificial cerebrospinal fluid (ACSF) containing 120 mM NaCl, 1.25 mM $NaH_2PO_4$, 3.5 mM KCl, 2.5 mM $CaCl_2$, 26 mM $NaHCO_3$, 1.3 mM $MgSO_4$, and 10 mM glucose. Field excitatory postsynaptic potentials (fEPSPs) at the hippocampal Schaffer collateral-CA1 (SC-CA1) pathway were triggered with a FHC stimulating microelectrode (*Li et al., 2022b*, *2023b*). The input-output (I/O), paired-pulse ratio (PPR) and long-term potential (LTP) were recorded. LTP was induced by 100 Hz stimulation. All stimulating pulses were 100 μs in duration. Data were acquired using a MultiClamp 700B amplifier and pCLAMP 10.0 software (Molecular Devices, San Jose, CA, USA). All chemicals used were purchased from Sigma.

## Immunostaining

Immunostaining was done according to our previous study (*Li et al., 2023b*; *Zhao et al., 2014*). Mice were deeply anesthetized with 3% isoflurane and then intracardially perfused with saline and 4% paraformaldehyde. The brains were then post-fixed in 4% paraformaldehyde for 4–6 h, dehydrated in 30% sucrose for 48 h, and sectioned into 40 μm slices. The CA1 region of the dorsal hippocampus were identified based on anatomical landmarks. Coronal sections of the dorsal hippocampus were taken from bregma −1.34 to −2.8 mm according to mouse brain atlas by *Paxinos & Franklin (2019)*, using the *corpus callosum* for orientation. The CA1 region appears as a crescent shape running parallel to the curved edge of the dorsal hippocampus, characterized by a thin band of densely packed pyramidal neurons. The border between the CA1 and CA3 regions is identifiable by changes in the density and orientation of pyramidal cells. These brain sections were incubated overnight at 4 °C with a primary anti-Iba1 antibody (1:400; Millipore, Burlington, MA, USA), followed by a 1-h incubation with FITC-conjugated donkey anti-mouse secondary antibody (Jackson Immuno Research, West Grove, PA, USA) at room temperature. Images were captured using a virtual slide microscope (VS120; Olympus, Tokyo, Japan) equipped with a 20× objective lens. Three representative coronal sections spaced equally along the anteroposterior axis of the dorsal hippocampus and three representative images taken per section were selected for quantification purposes. The density of microglia in the hippocampus was analyzed as $Iba1^+$ cells/$mm^2$ using ImageJ 1.5 software.

## Quantitative reverse transcription PCR (RT-qPCR)

Quantitative reverse transcription PCR was done according to our previous study (*Liu et al., 2024*). Mice were deeply anesthetized with 3% isoflurane. The hippocampal tissue

was freshly isolated and quickly collected in enzyme-free EP tube and stored at −80 °C until use. Total RNA was extracted using the PureLink RNA Mini Kit (Thermo Fisher Scientific, Waltham, MA, USA), followed by quantity and quality measurement with a NanoDrop 2000 spectrophotometer (Thermo Fisher Scientific, Waltham, MA, USA). Single-stranded cDNA was synthesized from 1 μg of RNA with SuperScript III reverse transcriptase (Invitrogen, Waltham, MA, USA). The reaction condition was as follows: 25 °C for 10 min, 50 °C for 30 min, 85 °C for 5 min. PCR-based quantification of transcripts was performed using a Thermal Cycler Dice Real Time System (Roche, Indianapolis, IN, USA) and QuantiFast SYBR Green PCR kit (Qiagen, Hilden, Germany). The PCR cycling parameters were as follows: initial denaturation at 95 °C for 10 min followed by 40 cycles of PCR reaction at 95 °C for 15 s, 60 °C for 1 min, and 72 °C for 1 min. The primer sequences used were as follows: *Iba-1-F GACGACCCTTCTTCGGG TTT, Iba-1-R GAGAGCCCACAATCTTGCCT; Il-6-F GCCTTCTTGGGACTGATGCT, Il-6-R GCCATTGCACAACTCTTTTCTCA; Cd68-F ACTTCGGGCCATGTTTCTCT, Cd68-R GGGGCTGGTAGGTTGATTGT; Tmem119-F AGCCTACTACCCATCCTCGT, Tmem119-R CTGGGTAGCAGCCAGAATGT; Bdnf-F GGCTGACACTTTTGAGCACGTC, Bdnf-R CTCCAAAGGCACTTGACTGCTG; Syp-F TCCTGCAGAACAAGTACCGAGA, Syp-R GGCCATCTTCACATCGGACAG.* $2^{-\Delta\Delta CT}$ method was used to normalize CT values against housekeeping gene *Gapdh* and to quantify relative expression. Triplicates were done for each sample. Assays were carried out in our own lab by investigators unaware of experimental design.

**Enzyme-linked immunosorbent assay (ELISA)**

Enzyme-linked immunosorbent assays (ELISAs) were done according to our previous study (*Guo et al., 2019*). The hippocampal tissue was freshly isolated and homogenized in 0.5 ml of ELISA buffer and centrifuged at 3,500× *g* for 10 min. The resulting supernatant was collected and stored at −80 °C until use. Concentrations of IL-6, Aβ1-40, and Aβ1-42 in the hippocampus were measured using mouse ELISA kits from Wuhan Colorful Gene Biological Technology Co., China. Absorbance values were measured at 450 nm using a 96-well microplate spectrophotometer, with triplicates performed for each sample.

**Euthanasia**

Humane endpoints were set to ensure animal welfare. Mice displaying severe distress (*e.g.*, over 20% weight loss, inability to eat or drink, severe lethargy, or unresponsiveness) were immediately euthanized using carbon dioxide ($CO_2$) asphyxiation for 5 min followed by cervical dislocation. No mice met these criteria. At the study's end, any remaining mice not needed for further research were humanely euthanized using $CO_2$ for 5 min to ensure minimal stress and suffering.

**Statistical analyses**

Data were expressed as mean ± S.E.M. and analyzed with GraphPad Prism 9.0 software. Sample size, ANOVAs or *t*-tests used for statistical analyses were described, with normal

distributions and equal variances confirmed before performing parametric statistical analyses. A significance level of $P < 0.05$ was considered statistically significant.

# RESULTS

## Adult depletion of microglia with PLX5622 causes mild spatial memory impairment

Consistent with previous reports (*López-Aranda et al., 2023*), we initially demonstrated that treatment with PLX5622, a highly selective brain-penetrant inhibitor of the CSF1R, for 3 weeks resulted in widespread depletion of microglia throughout the brain, including approximately 83% reduction in the hippocampus (Figs. 1A–1C; Unpaired $t$ test, PLX5622 chow *vs.* control chow, $P < 0.0001$). Under physiological conditions, microglial depletion in adult mice did not impact locomotor activity or anxiety-related behaviors (Figs. 1E and 1F), nor did it affect spatial learning (Fig. 1G) or swimming speed (Fig. 1I) in the WMW task. However, during the probe test, it was found that while both groups of mice were able to form long-term spatial memories (Fig. 1H; one sample $t$ test, $P < 0.001$ in comparison to random 50%), PLX5622-treated mice spent less time searching in the training quadrant than controls (Fig. 1H; Unpaired $t$ test, PLX5622 *vs.* CON, $t = 2.61$, $P < 0.05$), indicating mild impairment in spatial memory.

## Depletion or inactivation of microglia blocks LPS-induced spatial memory impairment

In previous studies, we have demonstrated that systemic administration of LPS (0.5 mg/kg) daily for 7 days impairs spatial learning and memory in C57BL6 mice (*Liu et al., 2024*). To investigate the effect of microglial depletion on LPS-induced memory impairment, we initiated PLX5622 treatment 2 weeks prior to LPS administration. MWM training started 24 h after the conclusion of LPS injection (*Jung et al., 2023*) (Fig. 2A). Our results indicate that microglial depletion prevented LPS-induced spatial learning and memory impairment (Figs. 2B–2D). Specifically, PLX5622-pretreated mice (PLX5662+LPS) exhibited reduced latency to platform compared to controls (CON+LPS) over the course of 7 training days (Fig. 2B; Two-way repeated measure ANOVA with Sidak's multiple comparisons test, $P < 0.05$ to $P < 0.001$ from training day 5 to day 7). During the probe test on day 7, PLX5622+LPS mice spent a significantly higher percentage of time navigating the training quadrant than CON+LPS mice (Fig. 2C; unpaired $t$ test, $t = 3.74$, $P < 0.01$). Both groups of mice displayed similar swimming speed during the probe test (Fig. 2D). Therefore, our findings demonstrate that adult depletion of microglia prevents LPS-induced impairment in both spatial learning and spatial memory.

Systemic administration of LPS has been reported to induce significant microglial activation in the hippocampus (*Zhao et al., 2019*; *Jung et al., 2023*). We sought to inhibit microglial activation by administering minocycline during LPS insult. Previous research has demonstrated that minocycline, the broad-spectrum tetracycline antibiotic with pleiotropic effects, is capable of inhibiting microglial activation and M1 polarization (*Schmidtner et al., 2019*; *Kobayashi et al., 2013*). We found that pretreatment with minocycline for 1 week replicated the blocking effect of PLX5622 on LPS-induced

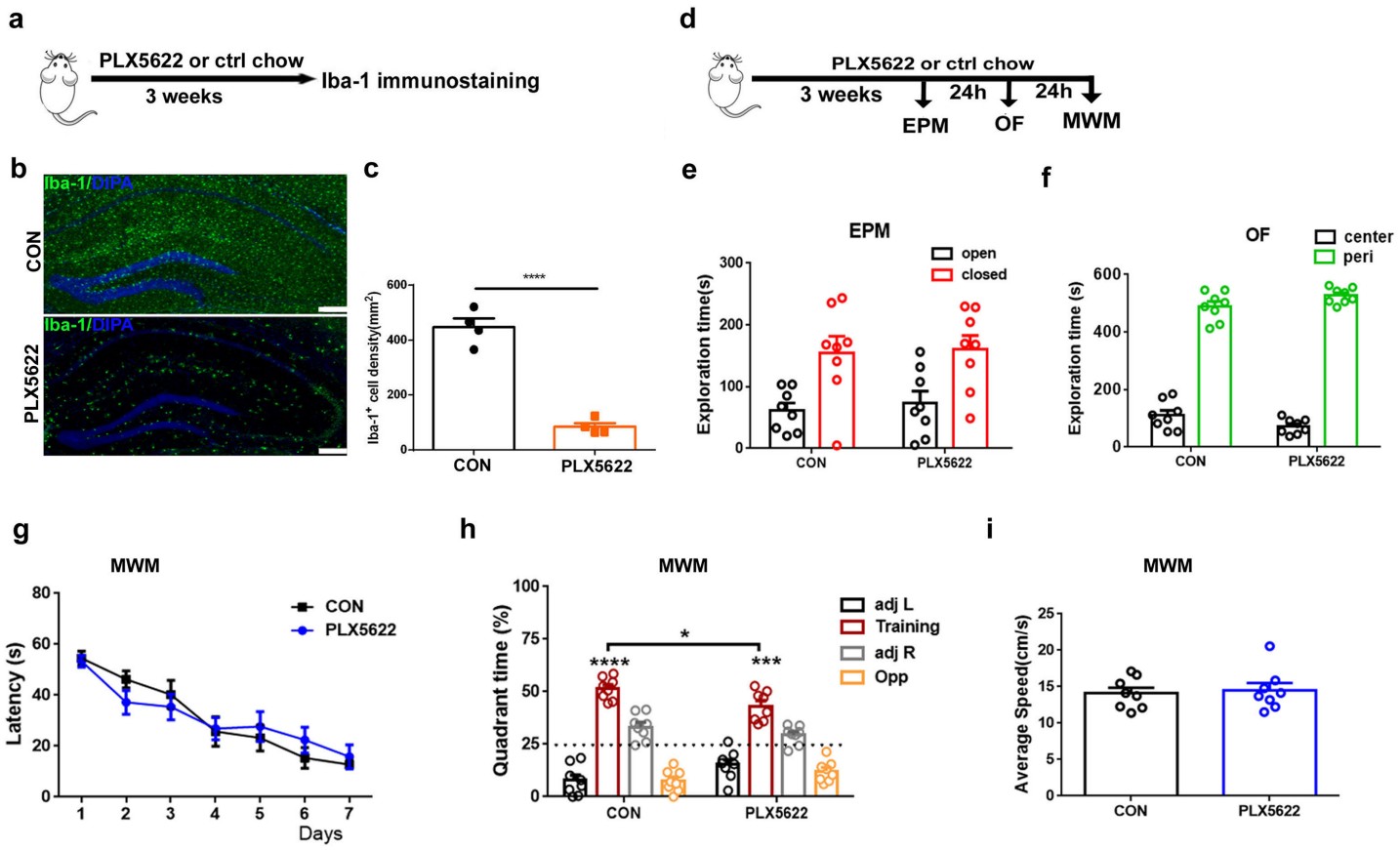

**Figure 1 Adult microglial depletion causes mild spatial memory impairment.** (A) Experimental design. (B) Representative images. Iba1 (green), DAPI (blue). Scale bar, 200 μm. (C) Iba1$^+$ cell density in the hippocampus. $n$ = 4 mice per group. (D) Behavioral experiment design. (E) EPM. (F) OF. (G–I) MWM. $n$ = eight mice per group. ****$P < 0.0001$, ***$P < 0.001$ or *$P < 0.05$ means significant difference.

cognitive dysfunction (Figs. 2E–2G). Similar to PLX5622, minocycline administration reduced latency to platform during MWM training (Fig. 2E; Two-way repeated measure ANOVA with Sidak's multiple comparisons test, minocycline+LPS *vs.* vehicle+LPS, $P < 0.05$ to $P < 0.01$), and increased training quadrant searching time during the probe test (Fig. 2F; Unpaired *t* test, minocycline+LPS *vs.* vehicle+LPS, $t = 3.40$, $P < 0.01$). Notably, minocycline treatment did not affect swimming speed (Fig. 2G). Overall, our results demonstrate that either depletion or inactivation of microglia can protect against LPS-induced spatial learning and memory impairment, highlighting the crucial role of proinflammatory activation of microglia in mediating LPS-induced cognitive deficits.

## PLX5622 treatment protects against LPS-induced synaptic dysfunction in the hippocampus

Next, we examined the possible impact of microglial depletion on LPS-induced synaptic dysfunction in hippocampal SC-CA1 pathway. Mice were fed with PLX5622 or control chow for 3 weeks prior to *ex vivo* field recordings in acute hippocampal slices isolated from LPS-insulted mice (Fig. 3A). Our results revealed that microglial depletion led to a slight

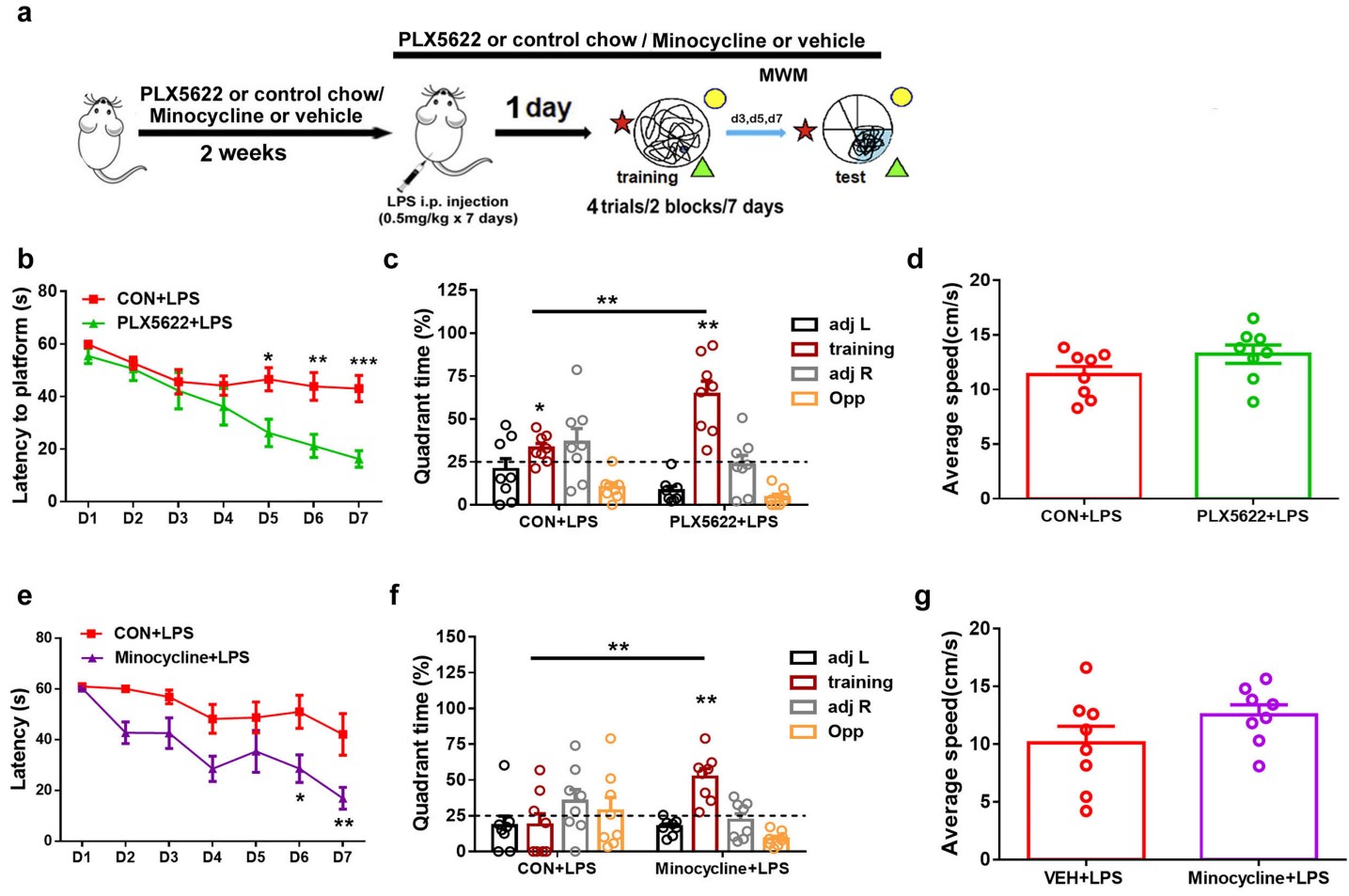

**Figure 2 Microglial depletion or inactivation protects against spatial memory deficit induced by LPS.** (A) Experimental design. (B–D) PLX5622 treatment. (B) Spatial learning curve, (C) the probe test, (D) swimming speed. (E–G) Minocycline treatment. *n* = eight mice per group. ***$P < 0.001$, **$P < 0.01$, or *$P < 0.05$ means significant difference.

increase in basal synaptic transmission (Fig. 3B; Two-way repeated measure ANOVA with Sidak's multiple comparisons test, PLX5622+LPS *vs.* CON+VEH, $P < 0.0001$ at 100 μA stimulation intensity) and a slight facilitation of paired-pulse ratio (Fig. 3C; PLX5622+LPS *vs.* CON+VEH, $P < 0.05$ at ISI of 50 ms) in SC-CA1 synapses. Notably, microglial depletion ameliorated LPS-induced LTP deficit (Figs. 3D–3F). Specifically, both the initial 5-min post-tetanic potentiation (PTP) and the last 20-min LTP were greater in PLX5622 +LPS mice than in CON+LPS mice (Fig. 3B; Unpaired *t* test, PTP: $t = 3.64$, $P < 0.01$; LTP: $t = 3.26$, $P < 0.01$). Therefore, our findings indicate that microglial depletion protects against LPS-induced synaptic dysfunction in the hippocampus, contributing to memory improvement.

## PLX5622 treatment suppresses LPS-induced proinflammatory activation of microglia in the hippocampus

Consistent with previous findings (*Skrzypczak-Wiercioch & Sałat, 2022*; *López-Aranda et al., 2023*), our RT-qPCR assays revealed upregulation of *Iba-1*, *Il-6*, and *cd68*, as well as

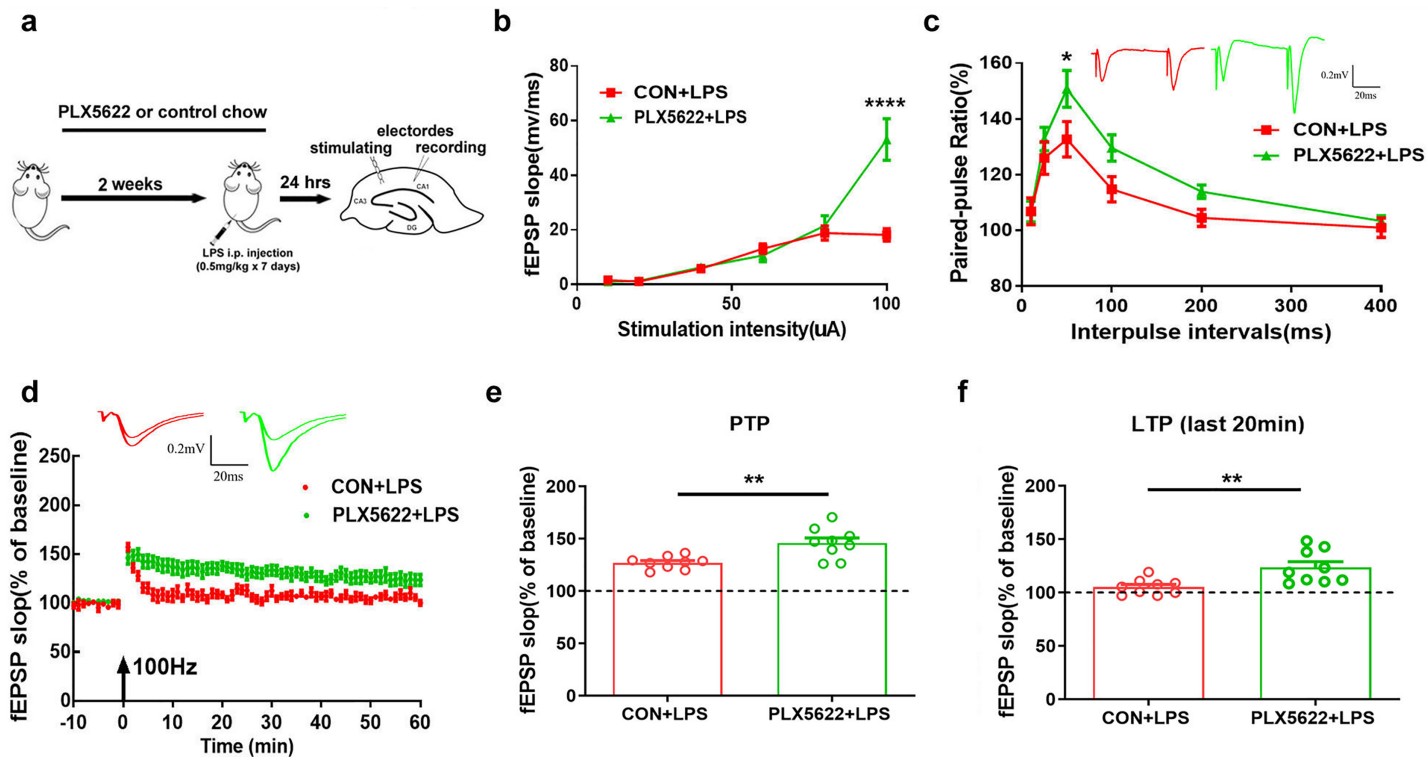

**Figure 3 Microglial depletion protects against LPS-induced synaptic dysfunction in SC-CA1 synapse.** (A) Experimental design. (B) I/O curve. (C) PPR. (D) LTP. (E) PTP comparison. (F) LTP comparison. $n$ = nine slices from five mice per group. ****$P$ < 0.0001, **$P$ < 0.01 or *$P$ < 0.05 means significant difference.

downregulation of *Tmem119* in the hippocampus following systemic LPS administration (Figs. 4A–4D; one sample *t* test, CON+LPS mice *vs.* control naïve mice, $P$ < 0.01), indicating proinflammatory activation of microglia. Additionally, we observed reduced expression of *Bdnf* and *Synaptophysin* (*Syp*) in the hippocampus of LPS-treated mice (Figs. 4E and 4F; one sample *t* test, CON+LPS *vs.* control naïve mice, $P$ < 0.01), suggesting synaptic deficit due to LPS insult. Notably, PLX5622 pretreatment inhibited LPS-induced microglial activation. Specifically, PLX5622+LPS mice exhibited significantly reduced expression of *Iba-1*, *Tmem119*, *Il-6*, and *cd68* in the hippocampus (Figs. 4A–4D; Unpaired *t* test, PLX5622+LPS *vs.* CON+VEH, $P$ < 0.01 to $P$ < 0.0001), along with a dramatic increase in *Syp* expression (Fig. 4F; Unpaired *t* test, $t$ = 8.01, $P$ < 0.0001). Moreover, our ELISA assays showed decreased levels of IL-6 (Fig. 4G) and Aβ1-40 (Fig. 4I) in the hippocampus of PLX5622+LPS mice (Unpaired *t* test, PLX5622+LPS *vs.* CON+LPS, IL-6: $t$ = 4.20, $P$ < 0.01; Aβ1-40: $t$ = 4.61, $P$ < 0.01), indicating that microglial depletion can mitigate LPS-induced microglial activation, which may contribute to improved synaptic and cognitive dysfunction.

## DISCUSSION

The CSF1R belongs to the class III transmembrane tyrosine kinase receptor family, predominantly expressed in microglia within the CNS (*Chitu et al., 2016*). CSF1R is

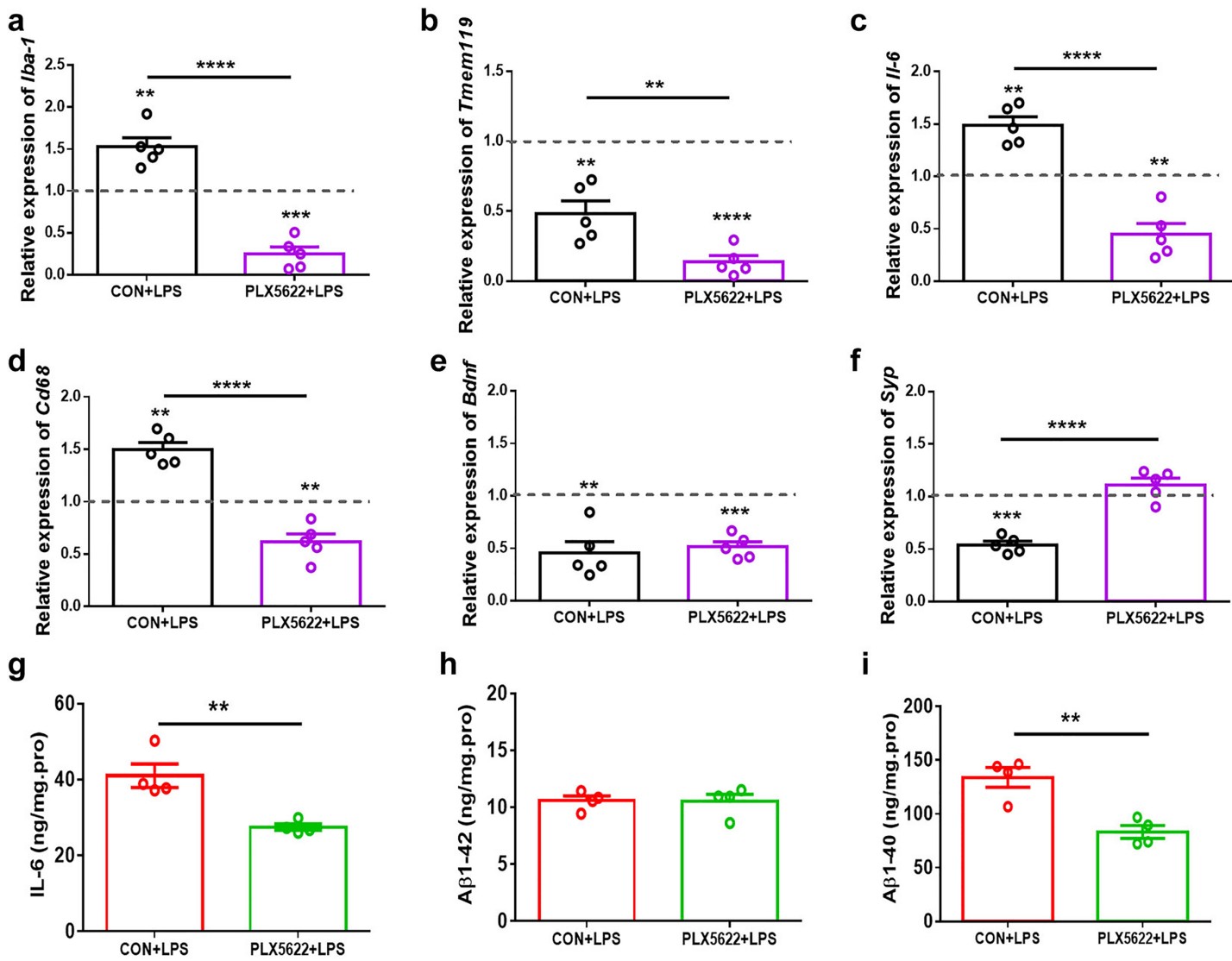

**Figure 4 PLX5622 treatment suppresses LPS-induced microglial activation in the hippocampus.** (A–F) RT-qPCR assays. *Iba-1* (A), *Tmem119* (B), *Il-6* (C), *Cd 68* (D), *Bdnf* (E), *Syp* (F). *n* = five mice per group. Dash line, mRNA expression in control mice without any treatment. (G–I) ELISA assays. IL-6 (G), Aβ1-42 (H), Aβ1-40 (I). *n* = four mice per group. ****P < 0.0001, ***P < 0.001, or **P < 0.01 means significant difference.

essential for the proliferation, differentiation, and survival of microglia (*Stanley et al., 1997*). Studies with CSF1R knockout mice have demonstrated complete absence of microglia at birth, accompanied by developmental defects (*Erblich et al., 2011*). Continuous CSF1/CSF1R or interleukin (IL)-34/CSF1R signaling is also required to maintain survival of microglia in adult brain (*Chitu et al., 2016*; *Bohlen et al., 2017*). The application of CSF1R inhibitors allows for time-dependent and reversible depletion of microglia in the CNS (*Elmore et al., 2014*). The inhibitors PLX3397 and PLX5622 are widely used to efficiently remove the majority of microglia from the brain (*Elmore et al., 2014*; *Spangenberg et al., 2019*). Consistent with previous reports (*Elmore et al., 2014*; *Spangenberg et al., 2019*; *Rice et al., 2017*), our study found that depleting microglia with

more selective CSFR1 inhibitor PLX5622 for 3 weeks did not affect locomotor activity or anxiety-like behavior. Therefore, we propose that the observed dual effects of microglial depletion on spatial learning and memory should be specific, highlighting the importance of microglia in regulating cognition under both physiological conditions and LPS-induced inflammation.

First, our findings demonstrate that adult microglial depletion for 3–4 weeks slightly impairs spatial memory, providing further evidence to support the notion that microglia in the healthy brain modulate memory strength and memory quality (*Cornell et al., 2022*). While many studies have investigated the effect of adult microglial depletion on spatial learning and memory under physiological conditions, results so far are rather divergent. For example, some research has reported no effect of microglial depletion for 3 weeks on spatial learning and memory in mice (*Elmore et al., 2014*; *Willis et al., 2020*), while others have found memory impairment in both young and aged rats after 3 weeks of microglial depletion (*Yegla et al., 2021*). In contrast, longer periods of depletion (up to 24 weeks) have been found to improve spatial learning and memory (*Elmore et al., 2014*; *Rice et al., 2015*; *Spangenberg et al., 2019*). The reason for this discrepancy is unclear; it may be due to differences in the duration of microglia depletion or variations in animal sex or age (*Basilico et al., 2022a*).

Secondly, we demonstrate that depleting adult microglia protects against LPS-induced dysfunction in both synaptic plasticity and spatial learning and memory. Previous studies have indicated that systemic LPS treatment promotes M1 activation while inhibiting M2 activation of microglia (*Zhao et al., 2019*). In our study, we found an increase in IL-6 expression in the hippocampus of LPS-treated mice, but no significant change in TNF-α or IL-1β. The discrepancy may be attributed to the timing of observations following LPS administration. We assessed cytokines expression in the mice hippocampus after completing MWM training and probe test, at a delayed time point rather than shortly after LPS injection. Importantly, we discovered that the upregulation of IL-6 induced by LPS coincided with increased expression of *Iba1*, *Cd68*, as well as decreased expression of *Tmem119*. These findings indicate the proinflammatory activation of microglia. More importantly, our results demonstrate that PLX5622 treatment not only reverses LPS-induced aberrant expression of *Il-6*, *Iba1*, and *Cd68* in the hippocampus, it also promotes synaptophysin expression, a marker protein in presynaptic vesicles (*Sheppard, Coleman & Durrant, 2019*). These changes could contribute to improved synaptic plasticity, including both PPR and LTP, and eventually lead to improved spatial learning and memory.

Furthermore, we confirm that inhibiting microglial activation and M1 polarization by minocycline mimics the protective effect of microglial depletion on LPS-induced cognitive dysfunction. Minocycline is a second-generation tetracycline antibiotic with high lipid solubility, allowing it to cross the BBB effectively (*Garrido-Mesa, Zarzuelo & Gálvez, 2013*). In the CNS, minocycline exhibits diverse pharmacological effects beyond its antibacterial properties. It primarily suppresses microglial activation by inhibiting several proinflammatory cytokines, including IL-1β, IL-6, CCL8, and CXCL4 (*Bergold, Furhang & Lawless, 2023*; *Bergold, 2016*). Minocycline mitigates neuroinflammation by

downregulating Notch signaling in microglia, inactivating MAPK pathway, and suppressing Nrf2-dependent antioxidation and NF-κB activity to reduce microglial M1 polarization (*Liang et al., 2023*; *Tian et al., 2017*). Altogether, we conclude that proinflammatory activation of microglia is determinant to LPS-induced memory impairment, and microglial depletion or inactivation is beneficial under persistent neuroinflammation conditions such as neurodegenerative diseases.

Chronic activation of microglia often leads to a sustained release of pro-inflammatory cytokines and reactive oxygen species, contributing to neuronal damage and synaptic dysfunction (*Leng & Edison, 2021*). The application of PLX5622 and minocycline can mitigate chronic activation of microglia, decrease overall inflammatory burden, restore homeostasis in the CNS, and protect neurons against damage or loss (*Šimončičová et al., 2022*; *Pang et al., 2012*). Supportively, pharmacological depletion or inactivation of microglia has been proven to reduce amyloid plaque load at early pathological stages of AD model mice (*Kater et al., 2023*). In addition, studies have indicated that microglial depletion or inactivation facilitates neuroprotective pathways, reduces abnormal synaptic pruning, increases dendritic spine density, and promotes neurogenesis (*Rice et al., 2015*; *Henry et al., 2020*; *Elmore et al., 2018*; *Zheng et al., 2022*). PLX3397 has been approved by the FDA for the treatment of tenosynovial giant cell tumor (TGCT), though it carries a black box warning due to the risk of severe fatal liver damage (*Spierenburg et al., 2022*). Clinical studies on PLX5622 are still limited. A safety evaluation in healthy adults (NCT01282684) has been completed, but the results have not yet been disclosed. Minocycline has been more extensively studied in human trials, particularly in psychiatric and neurodegenerative diseases, including major depressive disorder (MDD), multiple sclerosis (MS), and AD (*Howard et al., 2020*; *Möller et al., 2016*; *Hellmann-Regen et al., 2022*). Short-term (9 months) use of minocycline has been well tolerated (*Gordon et al., 2007*), but long-term (2 years), high-dose (400 mg) use may be less tolerable in elderly patients (*Howard et al., 2020*). Given the nonspecific effects of both minocycline and CSF1R inhibitors on peripheral immune cells, further clinical research is warranted to fully assess their safety and efficacy in CNS applications.

Inflammation-associated cognitive dysfunction is challenging, but there are several preventive and diagnostic measures that can help mitigate its detrimental effects. Recent studies have reported that healthy lifestyle such as anti-inflammatory diet, regular physical exercise, adequate sleep, and stress management helps to reduce inflammation and abnormal activation of microglia, and support cognitive health. Controlling chronic conditions like diabetes and obesity also helps to prevent inflammation-related cognitive decline (*Tan et al., 2023*). Blood tests, neuroimaging and cognitive assessments are useful in early detection of inflammation-associated cognitive dysfunction (*Patnode et al., 2020*). However, whether these measures can prevent or diminish neurodegeneration and associated memory impairment is still uncertain.

It is worth noting that microglial depletion is originally proposed as a tool to study microglial function and as a strategy to re-establish homeostasis in the diseased brain. However, it actually produces distinct effects on cognition ranging from protective to ineffective or even detrimental due to the high dynamics and heterogeneity of microglia *in*

*vivo* (*Basilico et al., 2022a*). Recent studies have shown that microglia depletion restores cognitive decline in LPS-insult mice (*Chen et al., 2023*), PD model mice (*Zhang et al., 2021*), aged 3xTg-AD model mice (*Dagher et al., 2015*), but not in 5XFAD-AD model mice (*Spangenberg et al., 2019*). New tools for manipulating microglia with more specificity and higher spatial and temporal resolution will become essential for future studies. Additionally, our study has several limitations that should be considered. The LPS-induced inflammation mouse model may not fully capture the complexity of human neuroinflammation conditions. Our focus on a specific timeframe for microglial depletion restricts the understanding of temporal aspects of treatment efficacy. Moreover, further studies are needed to illustrate the precise molecular and cellular mechanisms mediating the involvement of microglia in spatial learning and memory.

In conclusion, our findings provide new insight into the role and mechanism of microglial activation in LPS-induced spatial learning and memory impairment. Targeting microglia may provide a potential strategy for accurately treating inflammation-associated cognitive dysfunction.

## ABBREVIATIONS

| | |
|---|---|
| **Aβ** | Amyloid Beta |
| **ACSF** | Artificial Cerebrospinal Fluid |
| **AD** | Alzheimer's Disease |
| **ASD** | Autism Spectrum Disorder |
| **BBB** | Blood-Brain Barrier |
| **BDNF** | Brain-derived neurotrophic factor |
| **CCL8** | C-C Motif Chemokine Ligand 8 |
| **CD68** | Cluster of differentiation 68 |
| **CNS** | Central Nervous System |
| **CO$_2$** | Carbon dioxide |
| **CSF1R** | Colony-Stimulating Factor 1 Receptor |
| **CXCL4** | C-X-C Motif Chemokine Ligand 4 |
| **DAPI** | 4,6-diamidino-2-phenylindole |
| **EPM** | Elevated Plus Maze |
| **ELISA** | Enzyme-Linked Immunosorbent Assay |
| **fEPSP** | Field Excitatory Postsynaptic Potential |
| **GAPDH** | Glyceraldehyde-3-Phosphate Dehydrogenase |
| **Iba-1** | Ionized Calcium Binding Adapter Molecule 1 |
| **i.c.v.** | intracerebroventricular |
| **i.p.** | intraperitoneal |
| **IL-1β** | Interleukin-1 beta |
| **IL-6** | Interleukin-6 |
| **I/O** | curve Input/output curve |
| **LPS** | Lipopolysaccharides |
| **LTP** | Long-term potentiation |

| MAPK | Mitogen-Activated Protein Kinase |
|---|---|
| MS | Multiple sclerosis |
| MWM | Morris Water Maze |
| NF-κB | Nuclear Factor kappa-light-chain-enhancer of activated B cells |
| Nrf2 | Nuclear factor erythroid 2-related factor 2 |
| PD | Parkinson's Disease |
| PPR | Paired-pulse ratio |
| PTP | Post-tetanic potentiation |
| qPCR | Quantitative Polymerase Chain Reaction |
| RT-qPCR | Reverse Transcription-Quantitative Polymerase Chain Reaction |
| SC-CA1 | Schaffer collateral-CA1 Pathway |
| Syp | Synaptophysin |
| TGF-β | Transforming Growth Factor Beta |
| Tmem119 | Transmembrane protein 119 |
| TNF-α | Tumor Necrosis Factor Alpha |

### Funding

This work was supported by NNSFC (32071141, 32371211 to Yu Zhou), NSF of SD province (ZR2019ZD34 to Yu Zhou, ZR2023MH051 to Ming Yu), and NSF of Qingdao (23-2-1-188-zyyd-jch to Ming Yu). The funders had no role in study design, data collection and analysis, decision to publish, or preparation of the manuscript.

### Grant Disclosures

The following grant information was disclosed by the authors:
NNSFC: 32071141, 32371211.
NSF of SD province: ZR2019ZD34, ZR2023MH051.
NSF of Qingdao: 23-2-1-188-zyyd-jch.

### Competing Interests

The authors declare that they have no competing interests.

### Author Contributions

- Tao Zong analyzed the data, prepared figures and/or tables, authored or reviewed drafts of the article, and approved the final draft.
- Na Li performed the experiments, prepared figures and/or tables, and approved the final draft.
- Fubing Han performed the experiments, authored or reviewed drafts of the article, and approved the final draft.
- Junru Liu performed the experiments, prepared figures and/or tables, and approved the final draft.

- Mingru Deng performed the experiments, prepared figures and/or tables, and approved the final draft.
- Vincent Li analyzed the data, prepared figures and/or tables, authored or reviewed drafts of the article, and approved the final draft.
- Meng Zhang analyzed the data, authored or reviewed drafts of the article, and approved the final draft.
- Yu Zhou conceived and designed the experiments, authored or reviewed drafts of the article, and approved the final draft.
- Ming Yu conceived and designed the experiments, prepared figures and/or tables, and approved the final draft.

### Animal Ethics

The following information was supplied relating to ethical approvals (*i.e.*, approving body and any reference numbers):

The Chancellor's Animal Research Committee at Qingdao University.

### Data Availability

The raw measurements are available in the Supplemental File.

### Supplemental Information

Supplemental information for this article can be found online at http://dx.doi.org/10.7717/peerj.18552#supplemental-information.

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
