# Peer review of "Microglial depletion rescues spatial memory impairment caused by LPS administration in adult mice"

_PeerJ, doi:10.7717/peerj.18552_

## Round 0.1 · original submission · Major Revisions

Dear authors,

Thank you for your submission and for the work put into the manuscript titled "Microglial Depletion Rescues Spatial Memory Impairment Caused by LPS Administration in Adult Mice."
After careful consideration of the feedback from three reviewers, I have concluded that major revisions are necessary before the manuscript can be reconsidered for publication. Below, I highlight the key issues raised by the reviewers, identify specific areas that require special attention, and provide the reasoning behind the decision.

1) Experimental Justification:
Reviewer 1 and Reviewer 3 questioned the rationale behind the use of the C57BL/6J mice model and the decision to administer PLX5622 for three weeks. Please provide a more detailed justification for these choices, supported by relevant literature, to enhance the study’s credibility.

2) Discussion Expansion:
Reviewers 2 and 3 recommend expanding the discussion section, particularly to include a deeper analysis of microglia’s role in inflammation-related cognitive dysfunction and the mechanisms of action of PLX5622 and minocycline. Please also address their relevance to neurodegenerative diseases and previous studies that have explored these compounds in humans.

3) Materials and Methods Section:
The reviewers highlighted the need for a more detailed and referenced methodology. Specifically, please provide more citations and further clarify hippocampal tissue processing, immunostaining techniques, and the blinding protocols used in the study. These clarifications will add to the study’s methodological transparency.

4) Clarification of Abbreviations:
Both Reviewers 2 and 3 emphasized the importance of clearly defining abbreviations throughout the manuscript. Please ensure that a comprehensive list of abbreviations is provided at the end of the manuscript for clarity.

5) Study Limitations and Future Perspectives:
Reviewer 3 suggested discussing the limitations of your study and potential future research directions. This would help readers understand the study’s broader implications and how future work might address some of the current limitations.

6) Additional Suggestions
Grammar and Spelling: Reviewer 1 noted grammatical and spelling errors throughout the manuscript. We strongly encourage you to perform a thorough language check to improve the readability and overall professionalism of the text.

7) Reasoning for the Decision:
While the manuscript provides valuable insights into the role of microglia in LPS-induced spatial memory impairment, several critical aspects require further explanation and clarification. The absence of clear justifications for key experimental choices, incomplete methodological details, and gaps in the discussion of mechanisms weaken the impact of the current submission. Addressing these points will significantly strengthen the manuscript and enhance its potential for acceptance.

I look forward to receiving your revised manuscript and thank you for your continued efforts.

Sincerely,

Tiziano Balzano
Academic Editor
PeerJ Life & Environment

·

Basic reporting

The manuscript titled “Microglial depletion rescues spatial memory impairment caused by LPS administration in adult mice” provides important insights into the dual effects of microglia in memory function. The paper is well written and well structured.
The authors highlighted that either depletion or inactivation of microglia can protect against LPS-induced spatial learning and memory impairment, while adult microglial depletion for 3-4 weeks slightly impairs spatial memory. The references used are all relevant. Additionally, all the methods and procedures performed are appropriately described.

-Minor English editing is required, because there is several grammatical and spelling mistakes throughout the text, such as:
Line 53: Please change this “disrupted, resulting in memory impairment especially in the hippocampus” into “disrupted, especially in the hippocampus, resulting in memory impairment”.
Line 86: change “Mice was gently” into “Mice were gently”
Line 89: change “humane euthanasia” into “human euthanasia”

Experimental design

1. Why did the authors choose to work on C57BL/6J mice
2. Why did the authors exactly choose PLX5622 to administer during 3 weeks? Please add a reference

3. Behavioral assessments
The paper objective of the paper was to investigate the effect of adult microglial depletion on spatial learning and memory under both physiological conditions and LPS-induced neuroinflammation. Why the authors used Plus maze and Open field tests to evaluate motor function and anxiety?
4. Due to the limited brain tissue size, accurately pinpointing the specific locations within the hippocampal subfield, CA1, can be quite challenging. To achieve precise localization of this region, it is recommended that the author provides histological evidence using specific markers for tissue staining. This approach would validate that the designated area has been correctly identified and analyzed, ensuring the accuracy of the study's findings.

Validity of the findings

Data are well presented with accurate statistical analysis.
Conclusions are well stated and supported the results of the study

Additional comments

The manuscript titled “Microglial depletion rescues spatial memory impairment caused by LPS administration in adult mice” provides important insights into the role of microglia activation in LPS-induced impaired spatial learning and memory.

Reviewer 2 ·

Basic reporting

This manuscript provides highly valuable information regarding the importance of microglia and involvement of microglial immune-activation and cognitive deficit. The content is well-researched and makes a significant contribution to the current understanding of this field. I commend the authors for their extensive data set, the manuscript is clearly written in professional, unambiguous language. However, it has some flaws that, if addressed, would make the manuscript more appropriate.
1. Please discuss briefly the preventive and diagnostic measures microglia and inflammation-associated cognitive dysfunction.
2. In line 245, you shouldn’t begin the sentence with an abbreviation.
3. Please add the abbreviations of used names in full name before abbreviation in all fig. legends.

4. Please add list of abbreviations
5. The authors should declare that the research was conducted without any commercial or financial relationships that could lead to a conflict of interest.

Experimental design

no comment

Validity of the findings

no comment

Additional comments

no comment

Reviewer 3 ·

Basic reporting

The manuscript entitled "Microglial depletion rescues spatial memory impairment caused by LPS administration in adult mice" in which the authors evaluated the effect of depleting adult microglia on spatial learning and memory under both physiological conditions and lipopolysaccharide (LPS)-induced neuroinûammation. They found that microglial depletion by PLX5622 caused mild spatial memory impairment in mice under physiological conditions. Also, PLX5622 treatment suppressed LPS-induced neuroinûammation, microglial activation, and synaptic dysfunction.
The work is understandable and important. The scientific narrative is well structured and flows naturally from one idea to the next. However, this paper suffers from some shortcomings that if modified would make the manuscript suitable for publication in Journal of PeerJ.
Shortcomings:
1- In introduction section,
• The authors wrote “While studies have emphasized the importance of microglia in modulating cognitive functions 67 such as learning and memory in both healthy and diseased brains, the results are controversial”. Please add in brief some of these controversial studies demonstrating the gap of knowledge in this part and the need of performing this study.
• Please add in brief, the importance of PLX5622 and minocycline and their role on inactivation of microglia.
2- In Material and methods section,
• The whole section of Material and methods needs more citations, please add references to support the used techniques in your study including behavioral assessments, ELISA, immunostaining, PCR..... etc.
• Were the hippocampal tissue of mice cut into pieces to be used in different investigation such as immunostaining, PCR, etc? Please add this information.
• The authors wrote “At the study's end, any remaining mice not needed for further research 169 were humanely euthanized using CO2 to ensure minimal stress and suffering.” Please add the time of euthanasia.
• The information about the number of sections per specimen, and the number of photos taken per section is missing in immunostaining data. Please add.
• Please clarify the measures used for blinding in this study.
3- In discussion section,
• Please discuss in detail, the importance of PLX5622 and minocycline and their role on microglia including their mechanism of action.
• please discuss in brief, the effectiveness, safety and possible adverse effects of the PLX5622 and minocycline in human in previous studies?
• Please discuss in detail, how the depletion of microglia could have beneficial effects in chronic neuroinflammation and neurodegenerative diseases.
4- Please add the limitation of this study and your future perspectives.
5- In figure captions,
• Please add the abbreviations of used names in full name before abbreviation.
• Please justify the different number of samples in each figure legend.

Experimental design

2- In Material and methods section,
• The whole section of Material and methods needs more citations, please add references to support the used techniques in your study including behavioral assessments, ELISA, immunostaining, PCR..... etc.
• Were the hippocampal tissue of mice cut into pieces to be used in different investigation such as immunostaining, PCR, etc? Please add this information.
• The authors wrote “At the study's end, any remaining mice not needed for further research 169 were humanely euthanized using CO2 to ensure minimal stress and suffering.” Please add the time of euthanasia.
• The information about the number of sections per specimen, and the number of photos taken per section is missing in immunostaining data. Please add.
• Please clarify the measures used for blinding in this study.

Validity of the findings

The results are valid.

---

## Round 0.2 · accepted · Accept

After a thorough review, I confirm that you have addressed all the reviewers' comments effectively, incorporating the suggested changes and clarifying key points as needed.

I am pleased to inform you that this manuscript is now accepted and ready for publication.

·

Basic reporting

The manuscript titled “Microglial depletion rescues spatial memory impairment caused by LPS administration in adult mice” provides important insights into the dual effects of microglia in memory function. The paper is well written and well structured.
The authors highlighted that either depletion or inactivation of microglia can protect against LPS-induced spatial learning and memory impairment, while adult microglial depletion for 3-4 weeks slightly impairs spatial memory. The references used are all relevant. Additionally, all the methods and procedures performed are appropriately described.

Experimental design

The experimental design is clearly described, providing a thorough description of the methods and procedures adopted.

Validity of the findings

Data are well presented with accurate statistical analysis
Conclusions are well stated and supported the results of the study

Additional comments

This version of the manuscript titled “Microglial depletion rescues spatial memory impairment caused by LPS administration in adult mice” provides important insights into the role of microglia activation in LPS-induced impaired spatial learning and memory. The paper can be accepted in its current form.

Reviewer 2 ·

Basic reporting

There are no any comments now. The authors have modified all above comments.the manuscript becomes more appropriate now and we can accept it.

Experimental design

There are no any comments now. The authors have modified all above comments.the manuscript becomes more appropriate now and we can accept it.

Validity of the findings

There are no any comments now. The authors have modified all above comments.the manuscript becomes more appropriate now and we can accept it.

Reviewer 3 ·

Basic reporting

The manuscript entitled "Microglial depletion rescues spatial memory impairment caused by LPS administration in adult mice" in which the authors evaluated the effect of depleting adult microglia on spatial learning and memory under both physiological conditions and lipopolysaccharide (LPS)-induced neuroinûammation. They found that microglial depletion by PLX5622 caused mild spatial memory impairment in mice under physiological conditions. Also, PLX5622 treatment suppressed LPS-induced neuroinûammation, microglial activation, and synaptic dysfunction.
The revised manuscript is improved compared to prior revision. My comments were answered and explained by the authors. Therefore, I consider that the revised manuscript is acceptable and suitable for publication in Journal of PeerJ.

Experimental design

The manuscript entitled "Microglial depletion rescues spatial memory impairment caused by LPS administration in adult mice" in which the authors evaluated the effect of depleting adult microglia on spatial learning and memory under both physiological conditions and lipopolysaccharide (LPS)-induced neuroinûammation. They found that microglial depletion by PLX5622 caused mild spatial memory impairment in mice under physiological conditions. Also, PLX5622 treatment suppressed LPS-induced neuroinûammation, microglial activation, and synaptic dysfunction.
The revised manuscript is improved compared to prior revision. My comments were answered and explained by the authors. Therefore, I consider that the revised manuscript is acceptable and suitable for publication in Journal of PeerJ.

Validity of the findings

The manuscript entitled "Microglial depletion rescues spatial memory impairment caused by LPS administration in adult mice" in which the authors evaluated the effect of depleting adult microglia on spatial learning and memory under both physiological conditions and lipopolysaccharide (LPS)-induced neuroinûammation. They found that microglial depletion by PLX5622 caused mild spatial memory impairment in mice under physiological conditions. Also, PLX5622 treatment suppressed LPS-induced neuroinûammation, microglial activation, and synaptic dysfunction.
The revised manuscript is improved compared to prior revision. My comments were answered and explained by the authors. Therefore, I consider that the revised manuscript is acceptable and suitable for publication in Journal of PeerJ.